# Dual functions of Aire CARD multimerization in the transcriptional regulation of T cell tolerance

Yu-San Huoh[1,2], Bin Wu [1,2,5], Sehoon Park[2], Darren Yang[1,2,3], Kushagra Bansal[4,6], Emily Greenwald [2], Wesley P. Wong [1,2,3], Diane Mathis[4] & Sun Hur[1,2 ✉]

Aggregate-like biomolecular assemblies are emerging as new conformational states with functionality. Aire, a transcription factor essential for central T cell tolerance, forms large aggregate-like assemblies visualized as nuclear foci. Here we demonstrate that Aire utilizes its caspase activation recruitment domain (CARD) to form filamentous homo-multimers in vitro, and this assembly mediates foci formation and transcriptional activity. However, CARD-mediated multimerization also makes Aire susceptible to interaction with promyelocytic leukemia protein (PML) bodies, sites of many nuclear processes including protein quality control of nuclear aggregates. Several loss-of-function Aire mutants, including those causing autoimmune polyendocrine syndrome type-1, form foci with increased PML body association. Directing Aire to PML bodies impairs the transcriptional activity of Aire, while dispersing PML bodies with a viral antagonist restores this activity. Our study thus reveals a new regulatory role of PML bodies in Aire function, and highlights the interplay between nuclear aggregate-like assemblies and PML-mediated protein quality control.

[1] Department of Biological Chemistry and Molecular Pharmacology Blavatnik Institute at Harvard Medical School, Boston, MA 02115, USA. [2] Program in Cellular and Molecular Medicine Boston Children's Hospital, Boston, MA 02115, USA. [3] Wyss Institute for Biologically Inspired Engineering, Harvard University, Boston, MA 02115, USA. [4] Department of Immunology Blavatnik Institute at Harvard Medical School, Boston, MA 02115, USA. [5] Present address: NTU Institute of Structural Biology, School of Biological Sciences, Nanyang Technological University, Singapore 637551, Singapore. [6] Present address: Molecular Biology & Genetics Unit, Jawaharlal Nehru Centre for Advanced Scientific Research, Bangalore 560 064, India. ✉email: Sun.Hur@crystal.harvard.edu

Recent studies have shown that formation of aggregate-like assemblies is necessary for the function of a variety of proteins. Such functional aggregates include membrane-less molecular condensates (i.e., phase separation), and more structurally defined macromolecular assemblies that serve as signaling scaffolds inside the cell[1,2]. A common protein motif that mediates macromolecular assemblies in various vertebrate innate immune and cell death signaling pathways is the caspase activation recruitment domain (CARD)[3]. CARD domains mediate homo-typic protein:protein interactions and often form oligomers or filaments, which allows for rapid signal amplification in cytosolic signaling pathways. CARDs have also been identified in the Sp100 family of nuclear transcriptional regulators[4]. However, unlike the cytosolic CARDs, nuclear CARDs have been poorly characterized in terms of their biophysical properties and their functions in transcriptional regulation.

One Sp100 family member, whose role in adaptive immunity has been studied in depth, is the transcription factor Autoimmune Regulator (Aire). Aire is a multi-domain protein harboring a CARD (Fig. 1a), and plays a key role in establishing central tolerance in T cell immunity[5,6]. Aire is expressed predominantly in a subset of medullary thymic epithelial cells (mTECs) and induces the expression of thousands of peripheral tissue antigens (PTAs)[7]. Upregulated PTAs are then displayed on the surface of mTECs or cross-presented on dendritic cells for negative selection of auto-reactive T cells and positive selection of regulatory T cells[5,8,9]. Consistent with Aire's essential role in immune tolerance, mutations that impair proper Aire function cause a multi-organ autoimmune disease known as autoimmune polyendocrine syndrome type-1 (APS-1)[10].

Previous studies have shown that the molecular mechanism of how Aire induces gene expression may differ from those of conventional transcription factors. Aire binds only weakly and non-specifically to DNA, and no common sequence motif has been identified for Aire target genes[11,12]. Recent chromatin immunoprecipitation experiments suggest that Aire recognizes general epigenetic features and alters chromatin structure including super-enhancer and Polycomb-repressed regions, which may lead to indirect regulation of multiple target genes[13–16]. This idea is further supported by the stochastic nature of target gene expression at a single cell level[15,17,18], and the variance of Aire-dependent target genes depending on the cell type[19].

While the detailed mechanism for how Aire alters the chromatin landscape is as yet unclear, studies showed that Aire functions through the formation of large complexes involving both homo- and hetero-oligomerization with proteins involved in DNA repair, transcription, and mRNA processing[13,20,21]. Aire CARD is required for many of these interactions, including homo-oligomerization and binding to transcription regulators Brd4, Daxx, and SUMOylation pathway enzymes[22–25]. Cellular imaging studies have revealed that Aire forms nuclear foci[26,27], where presumably, these large complexes of Aire and its interaction partners reside. It has been shown that Aire foci formation is correlated with Aire transcriptional activity[4] and that these foci are distinct from other membrane-less nuclear granules characterized to date, such as promyelocytic leukemia protein (PML) bodies[26].

Here we report a combination of biochemical and cellular studies of Aire, which reveal the structural and functional properties of the Aire CARD-mediated homo-multimerization. We further show a previously unrecognized interaction between Aire assemblies and PML bodies, and characterize the role of this interaction in the pathogenesis of APS-1.

## Results

**Aire CARD forms filaments in vitro.** CARD domains are generally known to form helical assemblies, either in the form of small oligomers[28] or filamentous polymer[29,30]. In our effort to dissect the functions and structures of Aire CARD (Fig. 1a, Supplementary Fig. 1a), we expressed isolated mouse Aire (mAire) CARD in *E. coli* (Supplementary Fig. 1b). We found that purified mAire CARD forms filaments (Fig. 1b), and that the filaments are of non-amyloid type (i.e., no cross-β structure), as evidenced by Congo Red and thioflavin T staining assays (Fig. 1c, d). These observations are in agreement with previously characterized CARDs from cytosolic signaling molecules[29,30]. Interestingly, we also observed features of mAire CARD filament distinct from those of previously studied CARDs[29,30]. More specifically, mAire CARD filaments display at least two different thicknesses 15 nm and 10 nm, with 15 nm being the major population (Fig. 1b). Occasionally, the thinner filaments stem from thicker filaments, suggesting that thinner filaments may be protofilaments that integrate into thicker mature filaments. Such heterogeneity in mAire CARD filaments differ from previously characterized CARD filaments (e.g., MAVS and ASC CARDs), which form highly cooperative filaments of homogeneous thicknesses[29,30]. Our efforts to determine higher resolution structure of Aire CARD filaments were unsuccessful because of aggregation and distortion of the filaments on cryo-EM grids.

While cytosolic CARDs can form filaments with divergent helical symmetries, they utilize common surface areas in the conserved CARD fold[3]. We thus generated five mAire CARD variants with mutations in the putative filament interfaces based on a homology model of mAire CARD (Fig. 1e and Supplementary Fig. 1a, b). By negative-stain EM (Fig. 1f), we found only K51A/E52A displayed filament morphology similar to WT mAire CARD. Introducing mutations R16A/E18A and D35A/D37A completely impaired filament formation. K53A/E54A was limited to forming thin protofilaments and R70A/D71A formed tangled filamentous aggregates. Together, these observations suggest that mAire CARD forms filaments like cytosolic CARDs, but the assembly mechanism, filament interface contacts and/or interaction energetics may differ.

**CARD filament mediates transcription and foci formation.** We next asked whether our in vitro observation of CARD filament is relevant for nuclear foci formation and transcriptional activity of Aire. Studies of other filament forming proteins suggest that filamentous assembly can give rise to round foci in cells[31–34], presumably through tangling and higher-order assemblies of individual filaments. As such, direct visualization of the filament microstructure within the dense foci has been challenging. Thus, to understand how CARD filament formation mediates Aire foci formation and transcriptional activity, we examined the behavior of WT and Aire variants examined in Fig. 1f. We expressed Aire in human embryonic kidney 293 T cells and Aire-deficient human thymic epithelial 4D6 cells, two model cell lines that can recapitulate many aspects of Aire's function in mTECs, including transcriptional activity and interaction partners[13,20,25,35,36]. The transient expression conditions that we used closely emulate endogenous Aire's nuclear foci formation and the functional effects of patient mutations. Transcriptional activity of Aire was measured by the mRNA levels from a panel of genes known to be upregulated by Aire in 293 T or 4D6[20,25,37]. Our qPCR experiments (Fig. 1g and Supplementary Fig. 1c) combined with immunofluorescence microscopy (Fig. 1h and Supplementary Fig. 1d) suggest that filament formation likely plays an important role in Aire foci formation and transcriptional activity. Mutants defective in filament formation, i.e., R16A/E18A and D35A/D37A, showed a complete loss of transcriptional activity and Aire foci formation. On the other hand, K53A/E54A and R70A/D71A, which form thin filaments and tangled filaments, respectively

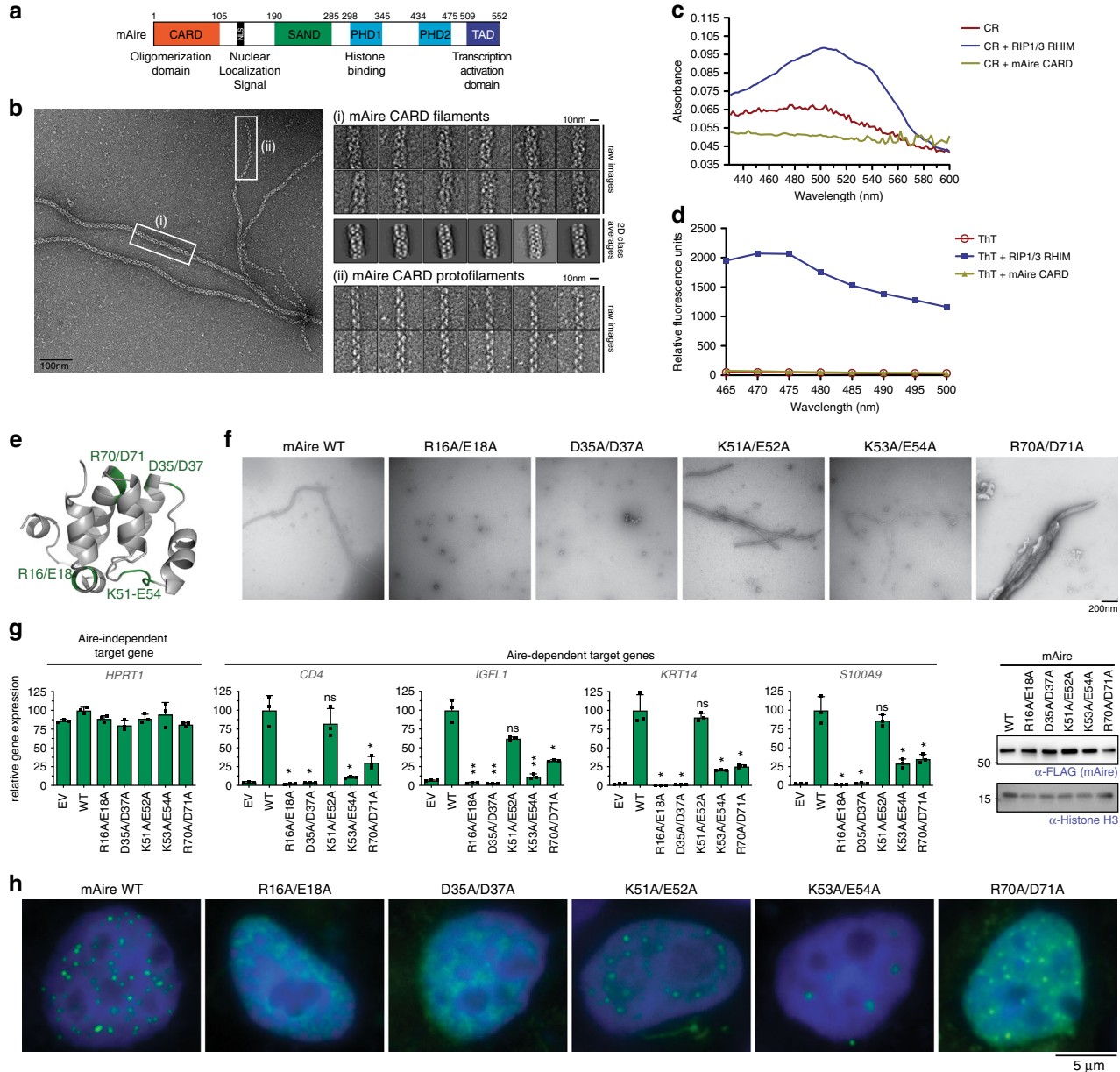

**Fig. 1 Filament assembly of Aire CARD mediates nuclear foci formation and transcriptional activity. a** Schematic of murine Aire domain architecture with previously characterized functions of individual domains. **b** Representative EM image of wild-type (WT) mAire CARD. While fully formed CARD filaments (i) were major species, thin protofilaments (ii) were also observed. Right, representative images of the two types of filaments. 2D class averages were also shown for fully formed CARD filaments. **c**, **d** Representative absorption spectra of Congo Red (CR, 15 μM) (**c**) or fluorescence emission spectra (excitation at 430 nm) of Thioflavin (ThT, 50 μM) (**d**) in the presence of WT mAire CARD filaments or known amyloids of RIP1/3-RHIM[70]. 5 μM monomeric concentration was used for both proteins. The experiments were performed two independent times. **e** 3D homology model of mAire CARD using the program FUGUE 2.01. Putative CARD:CARD interfaces were identified from the sequence alignment with other CARDs (Supplementary Fig. 1a) and are highlighted green. **f** Representative EM images of recombinant mAire CARD. WT protein and those with mutations in the putative CARD:CARD interfaces in (**e**) were compared. **g** Transcriptional activity of WT mAire and mutants, as measured by the relative mRNA levels of Aire-dependent genes (represented by *CD4*, *IGFL1*, *KRT14* and *S100A9*). An Aire-independent gene, *HPRT1*, was also examined as a negative control. Aire variants were transiently expressed (using 1.25 μg/ml DNA) in 293 T cells for 48 h prior to RT-qPCR analysis. Data are representative of at least three independent experiments and presented as mean ± s.d., n = 3. Right, western blot (WB) showing the expression levels of FLAG-tagged mAire compared to endogenous levels of Histone H3 (anti-H3). See Supplementary Fig. 1c for experiments in 4D6. *P*-values (two-tailed *t*-test) were calculated in comparison to WT mAire. \*p < 0.05; \*\*p < 0.01; p > 0.05 is not significant (ns). Exact *p*-values are provided in the Source Data File. **h** Representative fluorescence images of FLAG-tagged mAire in 293 T cells using anti-FLAG. See Supplementary Fig. 1d for experiments in 4D6.

(Fig. 1f), could still form nuclear foci, but had low transcriptional activity (Fig. 1g, h and Supplementary Fig. 1c, d).

To further test the importance of Aire CARD filament formation, we examined human Aire (hAire) CARD and five variants with APS-1 mutations L13P, T16M, L28P, A58V, and K83E[38,39] (Supplementary Fig. 1a). As with mAire CARD, we purified hAire CARD variants from *E. coli*, although purification required a refolding step (Supplementary Fig. 1e). We found that WT hAire CARD also forms filaments (Supplementary Fig. 1f). Under the same refolding condition, all five APS-1 mutants did

not form filaments (Supplementary Fig. 1f). All mutants have diminished transcriptional activity (Supplementary Fig. 1g). With the exception of A58V, all mutants also showed altered nuclear foci formation, displaying a diffuse pattern or forming enlarged aggregates (Supplementary Fig. 1h). Altogether, our analyses of mAire and hAire CARD variants collectively suggest that altered filament morphology or defect in filament formation is generally associated with loss of transcriptional activity along with perturbed nuclear foci formation.

**Aire homo-multimerization is required for function.** We next asked whether the precise structure of Aire filament is required or whether random multimerization is sufficient for the transcriptional activity of Aire. We swapped mAire CARD with 1–4 tandem repeats of FK506 binding protein[40] (FKBP1–4), generating FKBP1–4 N-terminally fused with ΔCARD of mAire (Fig. 2a). Individual FKBP domain can only homo-dimerize upon introduction of a chemical dimerizer, AP1903. However, tandem repeats of FKBP can form homo-multimers exponentially larger in size upon addition of AP1903. Note that the homo-multimerization of tandem repeats of FKBP is non-filamentous because the multimerization in this case is mediated by random association between two FKBP domains via AP1903. Deletion of CARD completely impairs the transcriptional activity of Aire (Fig. 2b and Supplementary Fig. 2a, b). FKBP1-ΔCARD and FKBP2-ΔCARD had little to no transcriptional activity with or without AP1903. Only the addition of 3–4 repeats of FKBP (FKBP3–4) was able to restore transcriptional activity in the presence of AP1903, albeit not to the same level as WT mAire

(Fig. 2b and Supplementary Fig. 2a, b). By cellular imaging, we found that the nuclear foci formation of the mAire variants largely correlated with their observed transcriptional activities. ΔCARD showed diffuse distribution (Fig. 2c), consistent with the importance of CARD in homo-multimerization. By contrast, FKBP4-ΔCARD showed nuclear foci only in the presence of AP1903 (Fig. 2c), indicating AP1903-dependent assembly of large homo-multimers. FKBP1/2-ΔCARD showed no such foci even in the presence of AP1903, while FKBP3-ΔCARD displayed fewer foci with diffuse background distribution (Fig. 2d). Altogether these results suggest that, unlike what has been previously proposed[23,35,41], dimerization or tetramerization of Aire is potentially insufficient for its functions; rather, large assembly is required. Furthermore, the transcriptional activity of FKBP4-ΔCARD suggests that filament structure per se is not essential for Aire, but this architecture satisfies the large homo-multimeric requirement.

**Aire mutants with increased PML association are inactive.** Our observations above raised the question why certain Aire CARD mutants, such as K53A/E54A, display low transcriptional activity while forming nuclear foci in cells and filaments (albeit with altered morphology) in vitro (Fig. 1f–h). To further investigate the effect of the multimerization domain on Aire function, we replaced Aire CARD by closely related CARDs of Sp100-like transcriptional regulators, such as Sp100, Sp110, and Sp140L[4]. Only Sp110-CARD swap expressed at levels comparable to WT Aire (Fig. 3a), and thus was chosen for transcriptional and imaging analysis. Intriguingly,

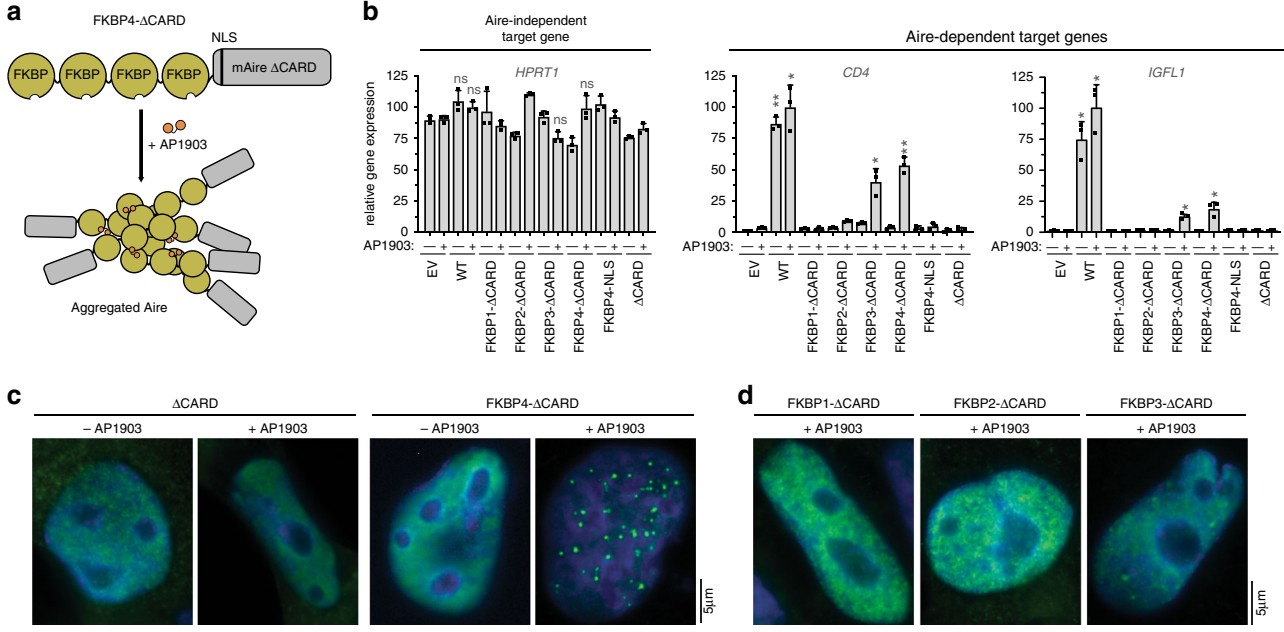

**Fig. 2 Chemically induced multimerization partially restores the transcriptional activity of AireΔCARD. a** Schematic of mAireΔCARD fused with four tandem repeats of FKBP (FKBP4-ΔCARD). Proteins with FKBP repeats are known to multimerize upon addition of chemical dimerizer AP1903[40]. **b** Transcriptional activity of ΔCARD fused with 1–4 tandem repeats of FKBP (FKBP1–4) in the presence and absence of AP1903. Fusion constructs were transiently expressed (using 1.25 μg/ml DNA) in 293 T cells and DMSO or AP1903 (5 μM) was added 24 h later. Cells were harvested 48 h after transfection, followed by RT-qPCR of the respective Aire-dependent genes (represented by *CD4* and *IGFL1*). The relative mRNA level of an Aire-independent gene, *HPRT1*, was also shown as a negative control. Data are representative of at least three independent experiments and presented as mean ± s.d., n = 3. See Supplementary Fig. 2a, b for additional target genes and western blot (WB) showing protein expression levels. *P*-values (two-tailed *t*-test) were calculated in comparison to empty vector (EV) + AP1903. *p < 0.05; **p < 0.01; p > 0.05 is not significant (ns). Exact *p*-values are provided in the Source Data File. **c**, **d** Representative fluorescence microscopy images of FLAG-tagged ΔCARD fusion variants in 4D6. Note that 4D6 cells were used as these cells are flatter than 293 T, allowing more robust analysis of Aire nuclear localization. Fusion constructs were transiently expressed and treated with DMSO or AP1903 as in (**b**). 48 h after transfection, cells were immunostained with anti-FLAG (mAire) and anti-PML. See Supplementary Fig. 2c, d for endogenous PML immunostaining.

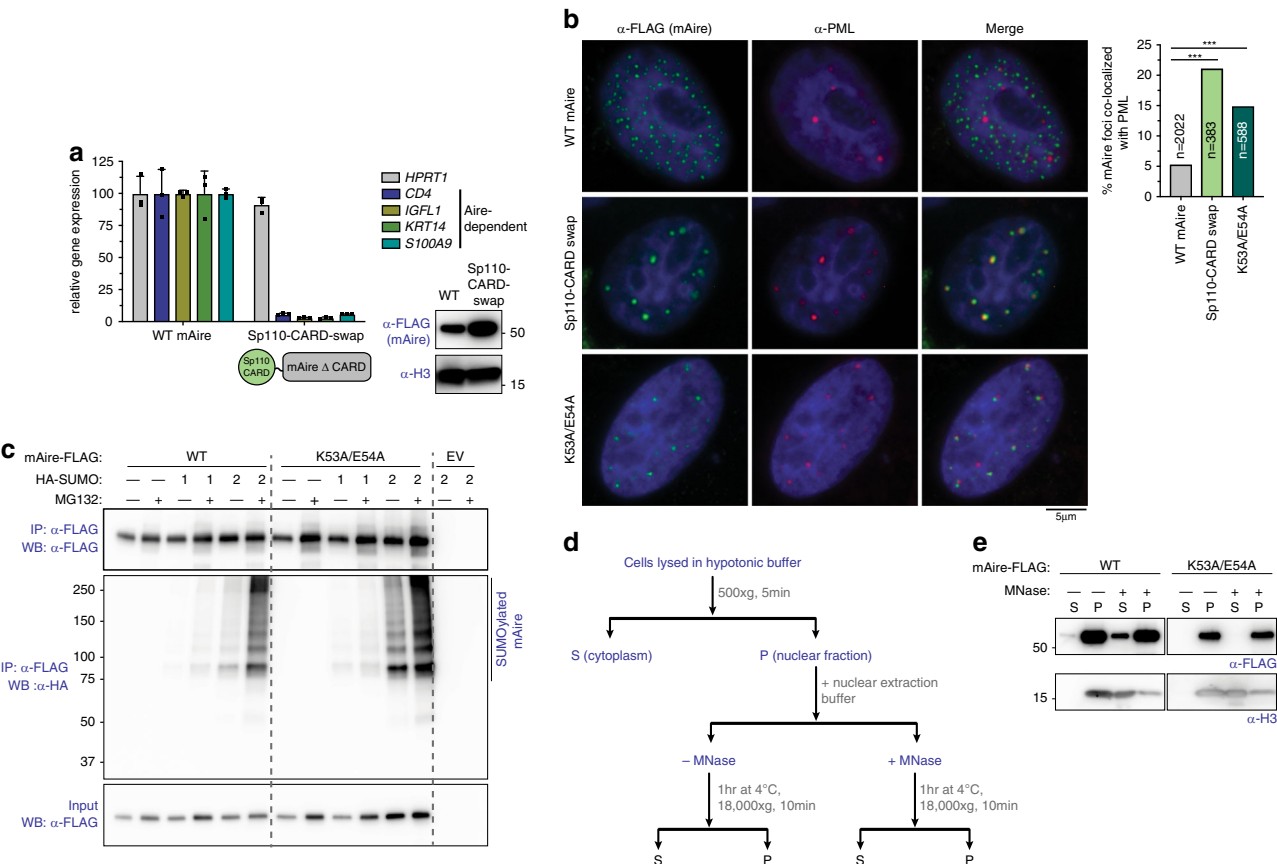

**Fig. 3 Aire mutants with increased PML association are impaired in transcriptional activity and are hyper-SUMOylated. a** Transcriptional activity of mAire and Sp110-CARD swap, where Aire CARD was swapped with Sp110 CARD. Experiments were performed as in Fig. 1g and presented as mean ± s.d., $n = 3$. Bottom right, WB of mAire and histone H3. **b** Representative fluorescence microscopy images of FLAG-tagged WT, Sp110-CARD swap and K53A/E54A mAire variants in 4D6 cells. Note that 4D6 cells were used as these cells show more distinct PML bodies and are flatter than 293 T, allowing more robust analysis of Aire foci and their endogenous PML body localization. Cells were immunostained with anti-FLAG (mAire) and anti-PML. Right, quantification of the Aire foci colocalized with endogenous PML bodies from automated image analysis (see Supplementary Fig. 3a, b for the definition of colocalization). $n =$ number of Aire foci examined per sample. Statistical significance comparisons were calculated using a two-tailed Student's $t$-test for two population proportions where each population consists of all individual Aire foci examined. ***$p = 5.57e$-$26$ and $6.46e$-$15$ for Sp110-CARD swap and K53A/E54A, respectively. See also Supplementary Fig. 3c for FLAG-tagged hAire stably expressed in 4D6 cells. **c** SUMO modification analysis of WT mAire and K53A/E54A. FLAG-tagged mAire was co-expressed with HA-SUMO1 or -SUMO2 in 293 T cells. Cells were treated with MG132 (10 μM) for 24 h before harvesting. mAire proteins were immunoprecipitated (IPed) using anti-FLAG beads under semi-denaturing condition and analyzed by anti-HA WB. **d** Schematic of chromatin fractionation analysis of Aire. 293 T cells were transfected with mAire expressing plasmids for 48 h before harvesting. Solubility of Aire and chromatin (as measured by Histone H3) before and after MNase treatment was analyzed by WB. **e** Chromatin fractionation analysis of mAire WT and K53A/E54A. Experiments were performed as in (**d**).

Sp110-CARD swap was transcriptionally inactive, despite the fact that it formed nuclear foci (Fig. 3a, b).

Sp110 along with other Sp100 family members, have been previously shown to colocalize with PML bodies, which are membrane-less nuclear structures often associated with transcriptional suppression[42,43]. We thus asked whether Sp110-CARD swap also colocalizes with PML bodies. Anti-PML immunofluorescence microscopy showed that WT Aire foci predominantly do not colocalize with PML bodies both in transient and stable expression systems (Fig. 3b and Supplementary Fig. 3a–c), consistent with previous observations in mTECs and model cell lines[26,44,45]. By contrast, Sp110-CARD swap showed increased association with PML bodies (Fig. 3b), as measured by the fraction of Aire foci with at least 50% shared area with PML bodies (Supplementary Fig. 3b). Intriguingly, nuclear foci of mAire K53A/E54A also have higher propensity to associate with PML bodies compared to WT Aire (Fig. 3b), while FKBP4-ΔCARD foci do not (Supplementary Fig. 2c). These results

suggest that Aire mutants with increased PML association are impaired in transcriptional activation.

We next examined whether PML localization is accompanied by changes in other biochemical properties of Aire. Since PML body localization is often associated with SUMOylated proteins[46,47], we examined SUMOylation of mAire K53A/E54A. We co-expressed FLAG-tagged Aire with HA-tagged SUMO in 293 T and immunoprecipitated Aire with anti-FLAG beads after lysate denaturation (a requirement to remove Aire-interaction partners). We then analyzed the eluate by anti-HA immunoblotting. Among the three functional mammalian SUMO isoforms (SUMO1–3), we focused our analyses on SUMO1 and SUMO2 because SUMO2 and SUMO3 share 97% sequence identity and are thought to be interchangeable[48]. While no substantial difference in SUMO1 conjugation was observed between WT and K53A/E54A mAire, K53A/E54A showed increased SUMO2 conjugation than WT mAire (Fig. 3c). The difference was more evident in the presence of proteasome inhibitor, MG132, which

appears to stabilize SUMOylated Aire. This is consistent with the notion that SUMOylated proteins are often targeted for ubiquitin conjugation and proteasomal degradation[49,50].

To further examine whether increased association with PML is accompanied by altered interaction between Aire and chromatin (and hence the loss of transcriptional activity), we performed chromatin fractionation analysis (Fig. 3d). For this assay, we used a nuclear extraction buffer that has been previously shown to be gentle enough to preserve Aire-interaction partners, and upon addition of micrococcal nuclease (MNase), would solubilize Aire along with its associated chromatin[20]. Indeed, chromatin (inferred by immunoblotting for histone H3) exclusively partitions in the insoluble fraction of nuclear extract until the addition of MNase, which frees chromatin into the soluble fraction (Fig. 3e). Similar to H3 partitioning, Aire remains mostly in the insoluble nuclear fraction until chromatin is solubilized with MNase treatment, which releases a portion of Aire. Intriguingly, MNase-dependent solubility enhancement was impaired for K53A/E54A mAire (Fig. 3e). These results together suggest that changes in the CARD domain can alter PML body association, SUMOylation, chromatin association and transcriptional activity of Aire.

**PML body localization can regulate Aire activity**. PML bodies are linked to not only transcriptional suppression, but also protein quality control of nuclear aggregates[51]. We therefore asked whether PML association actually contributes to the loss of transcriptional activities, or whether increased association with PML bodies is simply a consequence of Aire protein mis-folding due to mutations. To address this question, we first examined whether directing mAire to PML is sufficient to suppress mAire activity. Since SUMO-mediated protein:protein interactions are known to drive PML body formation[47,52], we fused Aire with the tandem repeats of SUMO interaction motif (SIM) of RNF4[53], a known component of PML bodies. As expected, SIM-mAire forms nuclear foci overlapping with PML bodies (Fig. 4a). The transcriptional activity of SIM-mAire was much lower than WT (Fig. 4b). These results suggest that PML colocalization can cause loss-of-function of Aire even without any mutation in the Aire protein itself.

To further examine the impact of PML localization on Aire activity, we next co-expressed WT mAire together with PML-localizing SIM-mAire. We found that the co-expression redirects WT mAire to PML bodies (Fig. 4c) presumably through Aire:Aire interactions, and impaired the transcriptional activity of WT mAire (Fig. 4d and Supplementary Fig. 4a). While SIM-ΔCARD alone showed more distributive pattern than SIM-Aire, part of it also formed foci that colocalized with PML bodies (Supplementary Fig. 4b). Intriguingly, WT mAire co-expression with SIM-ΔCARD also increased PML localization of WT Aire and partially suppressed the transcription activity of WT mAire (Fig. 4d and Supplementary Fig. 4b). This result suggests that CARD may not be the only domain mediating Aire homo-multimerizaiton. Regardless of the specific mechanism of homo-multimerization, these results collectively demonstrate that a PML colocalizing mutant can exert a dominant negative effect by altering Aire localization, further supporting the role of PML bodies in regulating Aire activity.

We next investigated whether disrupting PML bodies could rescue the negative effect of the PML-localizing Aire variants. Previous studies show that various viruses have evolved ways to antagonize PML's intrinsic immune function in suppressing viral gene transcription[54]. One such example is cytomegalovirus (CMV) intermediate-early protein IE1. Although IE1 has been implicated in various functions to counteract host immune

responses, IE1 in isolation can directly bind PML, prevent its SUMO conjugation activity, and disperse PML bodies[55–57]. Consistent with previous reports, we found that IE1 disperses PML bodies (Supplementary Fig. 4c). However, IE1 did not affect nuclear foci formation of WT mAire and the two mutants with increased association with PML bodies, K53A/E54A and SIM-mAire (Fig. 4e and Supplementary Fig. 4d). The transcriptional activity of K53A/E54A increased with IE1 and was fully restored beyond that of WT mAire without IE1 (Fig. 4f). The transcriptional activity of SIM-mAire also increased with IE1, albeit not to the same level as K53A/E54A (Fig. 4f). While we cannot exclude the possibility that IE1 affects Aire's activity independent of PML, these observations are consistent with the model that PML localization contributes to loss of function of Aire variants. Furthermore, the data undermine the notion that PML localization is a simple consequence of protein mis-folding.

Intriguingly, we also observed an increase in transcriptional activity of WT mAire upon co-expression with IE1, although the fold change was not as great as K53A/E54A or SIM-mAire (Fig. 4f). This raised the question whether WT mAire is also subject to PML-mediated regulation through potentially transient interaction with PML bodies. In fact, we noticed that WT Aire undergoes transient SUMOylation that becomes stabilized in the presence of MG132 (Fig. 3c). SUMOylation of Aire is dependent on CARD (Supplementary Fig. 4e), suggesting that CARD-mediated aggregate-like assembly of Aire is responsible. Furthermore, MG132-mediated stabilization of SUMOylated Aire was accompanied by increased PML localization and impaired transcriptional activity of WT Aire (Supplementary Fig. 4f, g). This suggests that even WT Aire may normally transit through PML bodies for quality control or degradation. While more detailed investigation is necessary to fully understand the effect of IE1, these results collectively support the model that PML bodies not only affect loss-of-function Aire mutants, but also WT protein.

**CARD is not the sole determinant for Aire foci localization**. Given the role of CARD in Aire multimerization and the impact of CARD mutations in PML colocalization so far, we next asked whether the location of Aire foci is solely determined by CARD. If so, one would predict that intact Aire CARD displays similar localization as WT Aire. We found that mAire aa 1–174, which harbors CARD and the nuclear localization signal (NLS) has very low expression levels, so we fused this to monomeric GFP (mGFP) (Fig. 5a). We found that CARD-mGFP colocalized with PML bodies (Fig. 5a), and also formed MNase-insensitive aggregates (Fig. 5b). Because of the previous report showing that mis-folded protein aggregates can colocalize with PML bodies[51], we asked whether CARD-mGFP is somehow mis-folded. Since we were able to detect GFP fluorescence of CARD-mGFP (an implication of properly folded GFP), we were left to determine if CARD was mis-folded. Therefore, we co-expressed CARD-mGFP with WT mAire to see if there were still CARD: CARD interactions between the two constructs. We found that CARD-mGFP indeed colocalizes with WT mAire upon co-expression (Fig. 5c). As with SIM-mAire, co-expression with CARD-mGFP induced PML localization of WT mAire (Fig. 5c) and impairs the transcriptional activity of WT mAire (Fig. 5d). Thus, in spite of its proper folding, CARD-mGFP localizes at PML bodies. Furthermore, while intact CARD is necessary for proper localization of Aire foci, CARD is not the sole determinant.

**PHD1 domain also contributes to Aire foci localization**. We next asked which other domain(s) of Aire contributes to proper

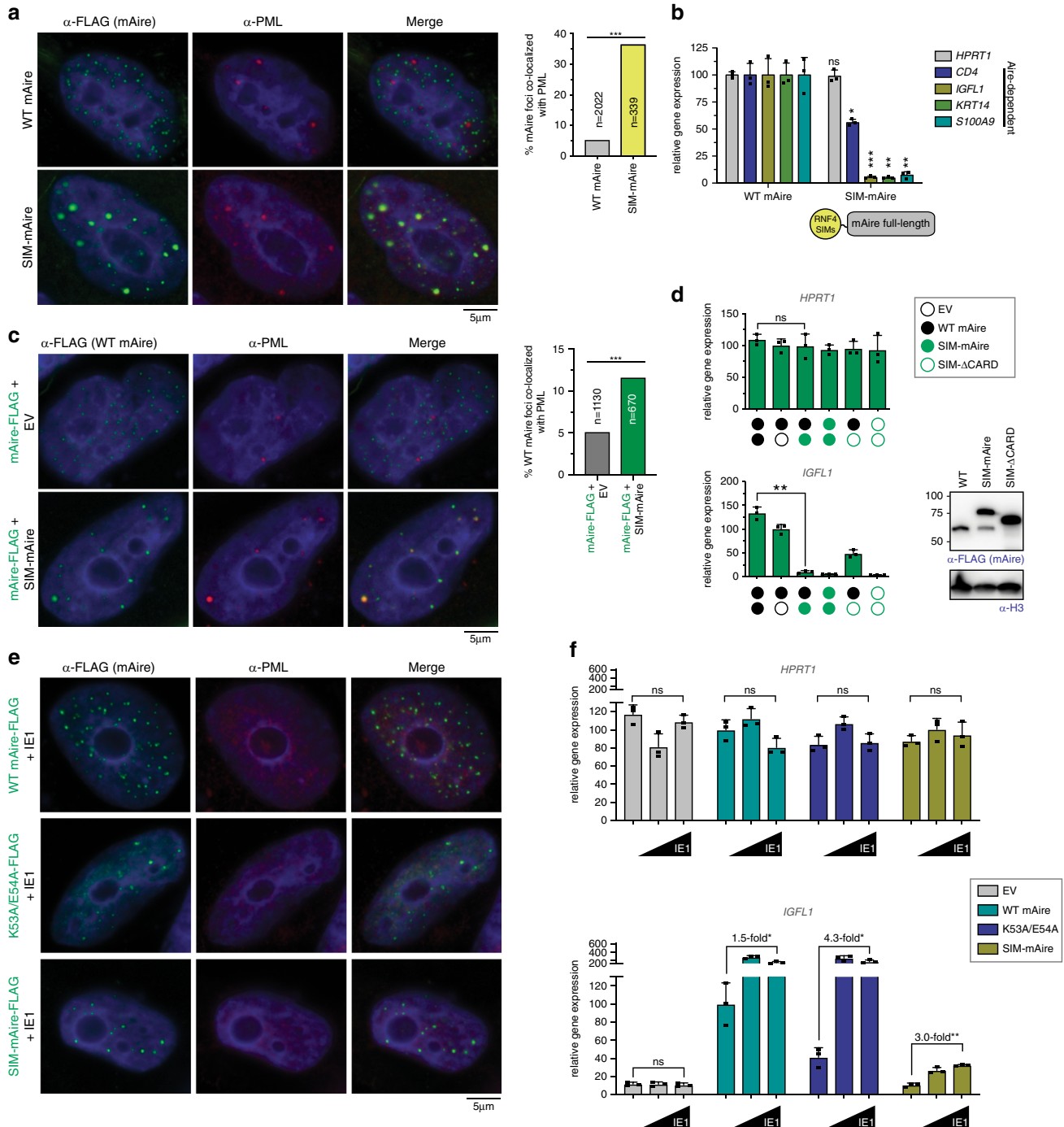

**Fig. 4 Directing Aire to PML bodies inhibits transcriptional activity, while dispersing PML bodies increases Aire activity. a** Representative fluorescence microscopy images of FLAG-tagged WT mAire and mAire N-terminally fused with SUMO interaction motifs (SIMs) from RNF4 (SIM-mAire). Right, quantification of Aire foci colocalized with endogenous PML bodies. $n$ = number of Aire foci examined per sample. ***$p$ (two-tailed Student's $t$-test) = 1.68e-71. **b** Transcriptional activity of mAire and SIM-mAire. Experiments are presented as mean ± s.d., $n$ = 3. $P$-values (two-tailed $t$-test) were calculated in comparison to WT mAire where *$p$ = 0.031; **$p$ = 0.0035 and 0.0014 for *KRT14* and *S100A9* respectively; ***$p$ = 0.0004; $p$ = 0.68 is not significant (ns). **c** Representative fluorescence microscopy images of WT mAire-FLAG with and without co-expression with SIM-mAire (no tag) in 4D6 cells. Right, quantification of the Aire foci colocalized with endogenous PML bodies as in (**a**). ***$p$ (two-tailed Student's $t$-test) = 4.39e-7. **d** Transcriptional activity of mAire (black circle) with and without co-expression of SIM-mAire (green circle) in 293 T cells. Each circle represents 0.6 µg/ml DNA transfected. Experiments are presented as mean ± s.d., $n$ = 3. See Supplementary Fig. 4a for additional Aire-dependent target genes. $P$-values (two-tailed $t$-test) were calculated in comparison to WT mAire where **$p$ = 0.002; $p$ = 0.88 is not significant (ns). **e** Representative fluorescence microscopy images of FLAG-tagged mAire co-expressed with CMV IE1 (0.5 µg/ml DNA each construct) in 4D6 cells. Cells were immunostained with anti-FLAG and anti-PML. See Supplementary Fig. 4c for the impact of IE1 on PML body dispersion. **f** Transcriptional activity of mAire, K53A/E54A and SIM-mAire (0.5 µg/ml DNA) with an increasing concentration of IE1 (0, 0.17, 0.5 µg/ml) in 293 T cells. Values are normalized against WT mAire without IE1. Experiments are presented as mean ± s.d., $n$ = 3. Fold-changes in transcriptional activities upon addition of IE1 are indicated. $P$-values (two-tailed $t$-test) were calculated in comparison to no IE1. *$p$ < 0.05; **$p$ < 0.01; $p$ > 0.05 is not significant (ns). Exact $p$-values are provided in the Source Data File.

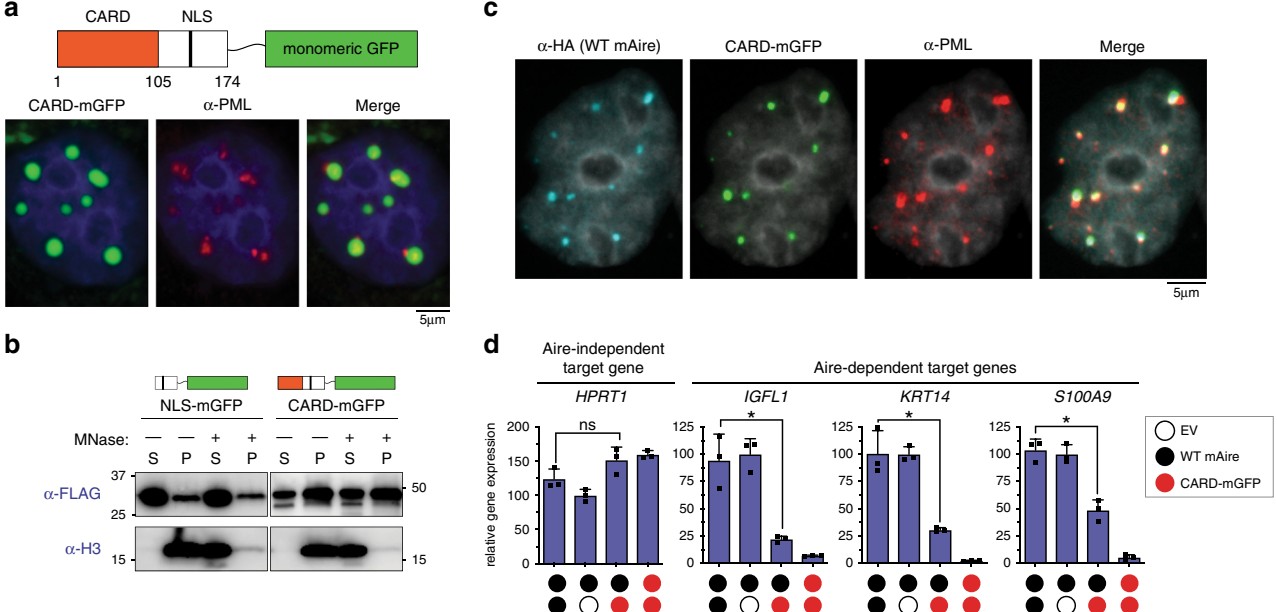

**Fig. 5 Isolated Aire CARD multimers associate with PML bodies. a** Representative fluorescence microscopy images (3 independent experiments) of mAire CARD fused with monomeric GFP (CARD-mGFP) transiently expressed in 4D6 cells. Note that mAire residues 1–174 containing both Aire CARD and nuclear localization signal (NLS) were used in the fusion construct. GFP fluorescence and anti-PML staining were used for imaging CARD-mGFP and PML bodies, respectively. **b** Chromatin fractionation analysis of NLS-mGFP (mAire aa 105–174 fused with mGFP) and CARD-mGFP. Experiments were performed as in Fig. 3d. **c** Representative fluorescence microscopy images (two independent experiments of CARD-mGFP and HA-tagged mAire upon their co-expression in 4D6 cells. GFP fluorescence was used for imaging CARD-mGFP and immunostained with anti-HA and anti-PML. **d** Transcriptional activity of mAire (black circle) and its changes upon co-expression with CARD-mGFP (red circle) in 293 T cells. Each circle represents 0.6 µg/ml DNA transfected. Experiments were performed as in Fig. 1g and presented as mean ± s.d., n = 3. P-values (two-tailed t-test) were calculated in comparison to WT mAire. *p < 0.05; p > 0.05 is not significant (ns). Exact p-values are provided in the Source Data File.

Aire foci localization. Since we observed that PML-localizing mutants could exert dominant negative effects on WT Aire, we turned our attention to the PHD1 domain, which was recently shown to harbor APS-1 mutations that display dominant negative effects[35]. PHD1 is also known to recognize unmethylated K4 on histone H3[11,12,58,59], but exactly why mutations in PHD1 exert a dominant negative effect has been unclear. We chose to examine C302Y and C311Y, two well-characterized loss-of-function dominant negative mutations[35] (Fig. 6a, b). We found C302Y and C311Y mutations increased the propensity of hAire to form PML-associated foci, both in stable and transient expression systems (Fig. 6c and Supplementary Fig. 5a, b). CARD is required for PML body localization and foci formation of C302Y (Supplementary Fig. 5c), suggesting that PML localization is not a simple consequence of mutation-induced protein mis-folding. These hAire variants also showed increased conjugation with SUMO2 and formed nuclear aggregates that are insensitive to MNase compared to WT hAire (Fig. 5d, e).

We next asked whether C302Y and C311Y also re-direct WT hAire to PML bodies, thereby exerting the dominant negative effect on WT hAire. Correlating with the dominant negative effects of C302Y and C311Y on WT hAire transcriptional activity (Fig. 7a), co-expression with C302Y or C311Y resulted in increased PML association of WT Aire foci (Fig. 7b) and decreased MNase-sensitivity of WT Aire fractionation (Supplementary Fig. 6a). To further investigate the role of PML localization in the dominant negative effect of these PHD1 mutants, we examined the impact of CMV IE1, as in Fig. 4f. The transcriptional activity of Aire WT with or without co-expression with C302Y variant increased in an IE1-dose dependent manner (Fig. 7c, d). The effect of IE1 was greater in the presence of C302Y than in its absence, in line with the increased PML association of

WT Aire upon co-expression with C302Y. The observed relief of the dominant negative effect of C302Y by IE1 is consistent with the model that PML localization can suppress Aire's transcriptional activity, but this suppression can be released upon PML body dissipation.

**Several dominant negative APS-1 mutants associate with PML.** To examine whether PML localization is more generally relevant for APS-1 pathogenesis, we examined additional APS-1 mutations. These include missense mutations in CARD, SAND, and PHD1[35,39,45,60,61] (Table 1). We also included a premature stop codon mutation, R257X[62], which truncates the Aire protein within the SAND domain. Most CARD mutants, with the exception of A58V, displayed distributive nuclear staining (Table 1 and Supplementary Fig. 1g, h, and 6b). By contrast, all SAND and PHD1 mutants examined in our study, including R257X, showed foci formation (Table 1 and Supplementary Fig. 6b). Interestingly, in addition to C302Y and C311Y in Figs. 6, 7, we found that G228W, R303P, and 257X showed increased PML association and exerted a dominant negative effect on co-expressed WT Aire (Table 1 and Supplementary Fig. 6b, c). Our observation of the dominant negative effect of G228W and R303P is consistent with previous reports[35,45]. PML localization and dominant negative effect of 257X have not been previously reported and are in line with what we observed of CARD-GFP (Fig. 5).

We also note that not all dominant negative mutants showed PML localization (Table 1). For example, R247C[60], previously reported to be dominant negative and confirmed to be so in our transcriptional activity assay (Supplementary Fig. 6c), showed nuclear foci that do not colocalize with PML bodies

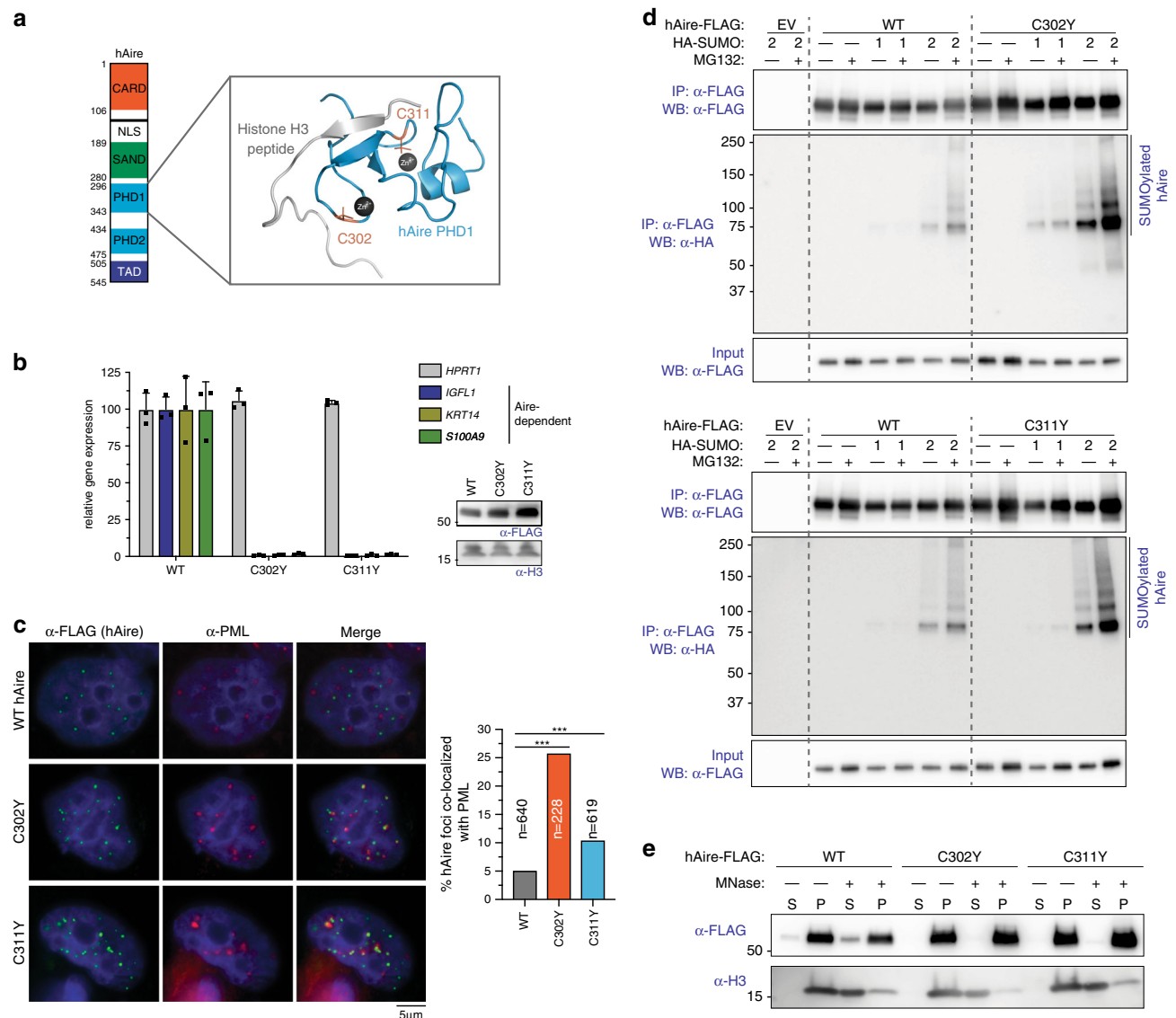

**Fig. 6 APS-1 mutations C302Y and C311Y increase PML association. a** 3D structure of Aire PHD1 domain bound to histone H3-derived peptide (PDB ID: 2KFT [https://doi.org/10.2210/pdb2KFT/pdb]). C302 and C311, which coordinate $Zn^{2+}$ ions, are mutated in APS-1 patients (highlighted in orange). **b** Transcriptional activity of hAire WT, C302Y and C311Y. Experiments were performed as in Fig. 1g and are presented as mean ± s.d., $n = 3$. **c** Representative fluorescence microscopy images of hAire C302Y and C311Y variants in 4D6 cells. Right, quantitation of Aire foci colocalized with PML bodies. $n$ = number of Aire foci examined per sample. Statistical significance comparisons were calculated using a two-tailed Student's $t$-test for two population proportions where each population consists of all individual Aire foci examined. ***$p = 2.6e-18$ (C302Y) and 0.0004 (C311Y). **d** SUMOylation analysis of hAire WT, C302Y (top) and C311Y (bottom). Experiments were performed as in Fig. 3c. **e** Chromatin fractionation analysis of hAire WT, C302Y and C311Y. Experiments were performed as in Fig. 3d.

(Supplementary Fig. 6b). Consistent with the lack of PML localization, we did not find hyper-SUMOylation or MNase insensitivity for R247C (Supplementary Fig. 6d, e). Altogether, these results suggest that while PML localization does not explain all APS-1 mutants, it is a feature frequently associated with dominant negative loss-of-function mutants.

## Discussion

We here show that the transcription factor Aire utilizes its CARD domain to form filamentous assemblies in vitro and provide evidence supporting that Aire filament mediates its transcriptional function in cells. While Aire can also form cytoplasmic fibers upon overexpression[26,27,62], such structures were not observed in native mTECs[45] and we thus restricted our analyses

to nuclear foci formation and transcriptional activity. The filamentous architecture is uniquely suited for forming large homomultimers, the requirement for optimal transcriptional activity of Aire. Large homo-multimerization, however, appears to function as a double-edged sword as it inevitably makes Aire susceptible to PML body-mediated protein quality control. While nuclear foci formed by WT Aire minimally overlaps with PML bodies, certain loss-of-function (LOF) mutants of Aire show increased association with PML bodies, SUMOylation, and loss of chromosomal association, jointly leading to the loss of Aire transcriptional activity. It should be noted that, by virtue of forming large homomultimeric, aggregate-like assemblies, even WT Aire seems subject to PML body-mediated regulation, albeit not to the same degree as the LOF mutants. This is further evidenced by SUMOylation of WT Aire, PML localization of WT Aire in the

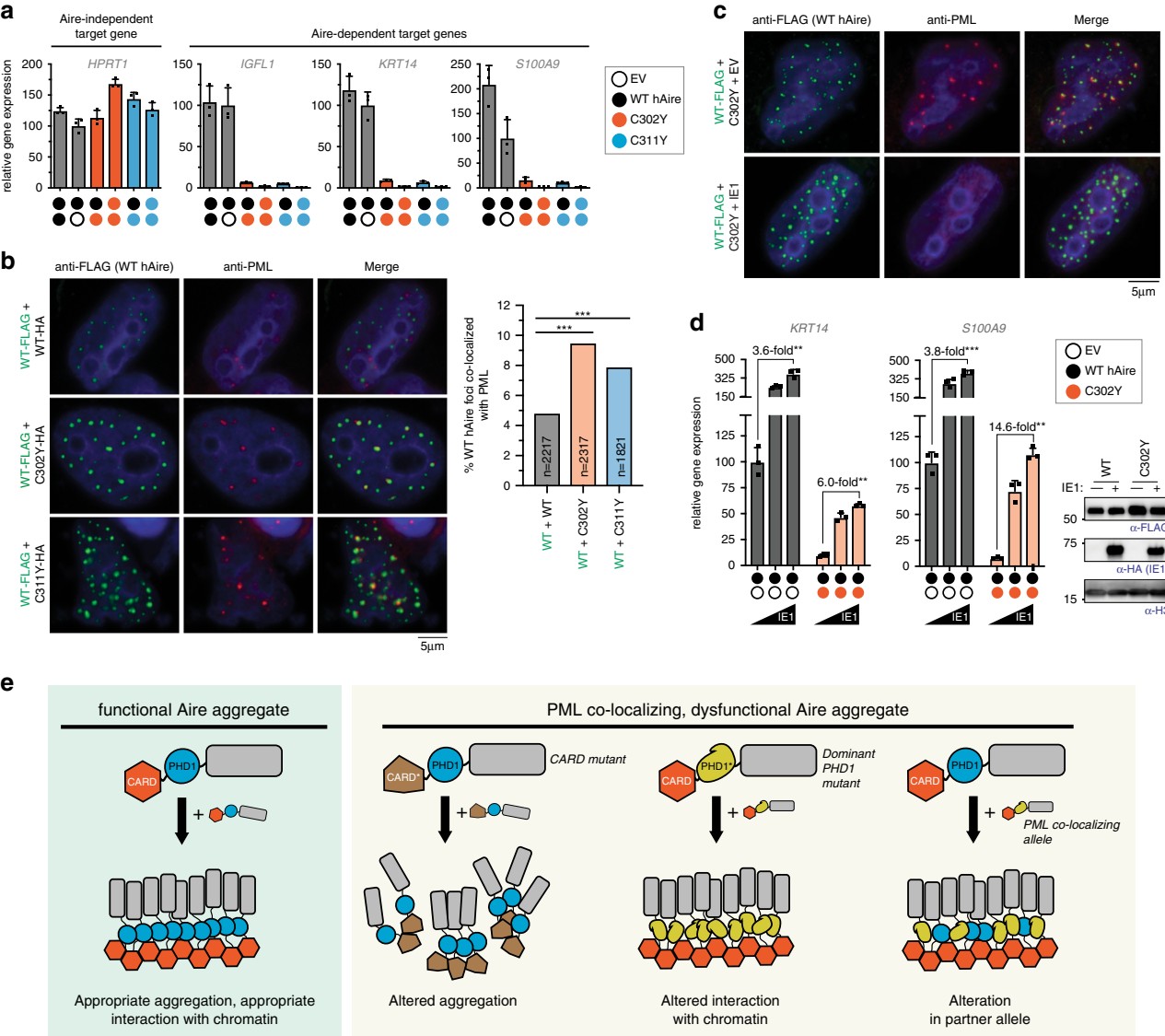

**Fig. 7 PML colocalization explains the dominant negative effect of C302Y and C311Y. a** Transcriptional activity of hAire (black circle) and its changes upon co-expression with C302Y (orange circle) and C311Y (blue circle) in 293 T cells. Each circle represents 0.6 μg/ml transfected DNA. Data are presented as mean ± s.d., n = 3. **b** Representative fluorescence microscopy images of hAire WT-FLAG co-expressed with hAire C302Y-HA (0.6 μg/ml DNA each) in 4D6 cells. Cells were immunostained with anti-FLAG (WT hAire) and anti-PML. Right, quantitation of Aire foci colocalized with PML bodies. n = number of Aire foci examined per sample. Statistical significance comparison was calculated using a two-tailed Student's t-test for two population proportions where each population consists of all individual Aire foci examined. ***p = 1.24e-9 (WT + C302Y) and 5.45e-05 (WT + C311Y). **c** Representative fluorescence microscopy images of hAire WT-FLAG co-expressed with hAire C302Y-HA (0.3 μg/ml DNA each) in the presence or absence of IE1-HA (0.5 μg/ml) in 4D6 cells. Cells were immunostained with anti-FLAG (WT hAire) and anti-PML. **d** Transcriptional activity of hAire WT (0.3 μg/ml) co-expressed with empty vector (0.3 μg/ml left panel) or hAire C302Y (0.3 μg/ml, right panel) and the impact of IE1 co-expression (0, 0.17, 0.5 μg/ml). Data are presented as mean ± s.d., n = 3. Fold-changes in transcriptional activities with IE1 are indicated. P-values (two-tailed t-test) were calculated in comparison to WT mAire. **p < 0.01; ***p < 0.001. Exact p-values are in the Source Data File. Right, western blot showing the expression levels of FLAG-tagged hAire variants co-expressed with or without IE1. **e** Model for PML body-mediated regulation of Aire function. Proper Aire function requires large homo-multimerization, but this property inevitably subjects Aire to PML body-associated protein quality control and transcriptional regulation. Note that PML body localization does not require protein mis-folding of Aire, and that the locations of Aire foci are not pre-defined. Instead, PML body localization can be induced by multiple factors, including altered multimerization property (through mutations in CARD), loss of chromatin interaction (through mutations in PHD1) or by interaction with PML colocalizing alleles.

presence of MG132, and the positive effect of IE1 on WT Aire transcriptional activity. These results together suggest a previously unrecognized regulatory role of PML bodies in Aire function, which is not only relevant to pathogenic LOF Aire mutants, but also WT protein.

Unlike the previous report where mis-folded protein aggregates such as Ataxin-1 are targeted to PML bodies[51], we found that

mis-folding is not required for PML localization of Aire (Fig. 5); rather, Aire association to PML appears to depend on multiple factors (Fig. 7e). Firstly, alteration in its multimerization property, either through mutations in CARD or swapping CARD domains, can increase PML body association. While most APS-1 mutations in CARD appears to lead to distributive nuclear staining, we found a few examples supporting the idea that altered CARD can

**Table 1 Nuclear localization and dominant negative effects of hAire APS-1 variants.**

| Domain mutated | hAire variant | Nuclear foci formation | % Aire foci colocalized with PML (significance) | Dominant negative effect |
|---|---|---|---|---|
| — | Wild-type | + | 5 | — |
| CARD | L13P | — | — | — |
| CARD | T16M | — | — | — |
| CARD | L28P | — | — | — |
| CARD | A58V | + | 6 (ns) | — |
| CARD | W78R | — | — | — |
| CARD | K83E | — | — | — |
| SAND | G228W | + | 14 (***) | + |
| SAND | R247C | + | <5 (ns) | + |
| PHD1 | E298K | + | 6 (ns) | + |
| PHD1 | C302Y | + | 26 (***) | + |
| PHD1 | R303P | + | 20 (***) | + |
| PHD1 | C311Y | + | 11 (***) | + |
| PHD1 | P326L | + | <5 (ns) | + |
| Premature truncation of SAND | 257X | + | 21 (***) | + |

P-values (two-tailed t-test) were calculated in comparison to WT hAire. ***p < 0.001; p > 0.05 is not significant (ns). Exact p-values are provided in the Source Data File. See Figs. 6c, 7a and Supplementary Fig. 1g, h and 6b, c for representative immunofluorescence images and transcriptional activities of the hAire variants.

induce PML localization without impairing foci formation. Secondly, mutations in PHD1 and the consequent loss of its interaction with chromatin, can also lead to an increased PML localization of Aire despite having an intact CARD. Along the same lines, isolated CARD and a truncated APS-1 mutant localize at PML bodies. Finally, even in the absence of any mutations, co-expression with PML-localizing mutants or treatment with MG132 can re-direct Aire to PML bodies. This suggests that proper localization of Aire foci not only requires intact Aire protein, but also appropriate partners and nuclear environment (Fig. 7e).

Why do certain Aire variants localize at PML bodies while others do not? Although the precise mechanism remains to be further investigated, we speculate that correct localization of Aire requires controlled multimerization of CARD at the right place and time. That is, Aire CARD may be normally suppressed (but not completely prevented) from having uncontrolled multimerization until it forms appropriate interactions with target chromatin sites via PHD1 (Supplementary Fig. 7a). As such, defect in PHD1 may prevent controlled multimerization, tilting the balance towards uncontrolled multimerization and PML localization (Supplementary Fig. 7b). Mutations in CARD may also lead to PML localization, if the mutation aberrantly releases suppression of uncontrolled multimerization (Supplementary Fig. 7c). For example, by swapping Aire CARD with Sp110 CARD, the inherent mechanism for suppressing uncontrolled multimerization would be incompatible with Sp110 CARD and therefore likely ineffective. Using this model, we can also explain how chemically induced homo-multimer of FKBP4-ΔCARD evades PML localization and is transcriptionally active (Fig. 2 and Supplementary Fig. 2). The addition of a chemical dimerizer to induce multimerization may mimic controlled multimerization of WT Aire as it provides the C-terminal portion of Aire sufficient time to form interactions with target chromatin before inducing multimerization.

How does increased PML association lead to transcriptional inactivation of Aire? PML bodies have been previously known to be enriched with transcriptional suppressors (e.g., Daxx or ATRX[63]), which offers an explanation for the observed association between PML localization and transcriptional suppression. At the quantitative level, however, our analysis suggests that a modest increase in PML colocalization (from ~5% in WT to ~15–30% in LOF mutants) is accompanied by a sizable loss of transcriptional activity. This non-linear effect on Aire activity was also seen with constructs such as SIM-Aire (~20–30%), for which full PML colocalization was expected. These observations led us to speculate that the non-linear effect reflects the limitation in our PML foci detection method or dynamic nature of the foci that cannot be fully analyzed by individual snap shots. Due to PML localization of fluorescently tagged Aire (Supplementary Fig. 7d, e), we were unable to perform live cell imaging of Aire-PML interaction. Future research is required to mechanistically dissect the link between PML body localization and transcriptional suppression.

Altogether, our study suggests the filament as the architecture of functional Aire homo-multimer and PML body as a previously unrecognized regulatory factor for Aire. While future research using mTECs and patient samples is necessary to further validate our model, our analyses using biochemical assays and model cell lines allow a direct comparison between WT Aire and mutants, and provide key insights into their intrinsic differences. Note that loss of functions and dominant negative effects of all APS-1 mutations, if not most, have been faithfully recapitulated in these model cell lines[35,45,60] (also see Table 1). Our study further suggests PML localization as a pathogenic mechanism for APS-1 and offers PML dispersion as a potentially new therapeutic strategy for treatment or prevention of APS-1. Finally, given the emerging role of large aggregate-like multimeric assemblies in transcriptional regulation[64–66], our findings on the PML-mediated transcriptional regulation may apply to a broad range of transcription factors beyond Aire.

## Methods

**Expression vectors.** Generation of pEGFP-N1-mAire-FLAG (EGFP gene removed), pCDNA3.1-hAire-FLAG, and -hAire-HA involved using restriction enzyme digestion and ligation to insert PCR-amplified cDNAs into respective vector backbones as described[11,60]. Point mutations within Aire plasmids were generated by using KAPA HiFi Hot start (Kapa Biosystems) or Phusion High Fidelity (New England Biolabs) DNA polymerases. The same mutagenesis strategy was used to introduce A206K point mutation in enhanced EGFP (monomeric EGFP, mGFP). In addition, a HindIII restriction site was introduced into pEGFP-N1-mAire-FLAG at aa104 to allow for ease of generating mAireΔCARD fusions. pEGFP-N1-mAire CARD-NLS (aa 1–173)-mEGFP and pEGFP-N1-mAire-NLS (aa 105–174)-mEGFP both have a 3XFLAG-TEV protease cleavage site linker between the mAire NLS and mEGFP. The constructs were made by first amplifying the Aire and mEGFP cDNAs separately, then introducing 3XFLAG-TEV linker to fuse the two amplified inserts together by overlap PCR. These overlap PCR products were then subcloned into pEGFP-N1 (EGFP gene in original vector removed). mAire aa 108–552 (mAire ΔCARD)-FLAG was subcloned into pEGFPN1. hAire aa 106–545 (hAire ΔCARD)-FLAG and aa 1–256 (257×)-FLAG were subcloned into pCDNA 3.1. The cDNA of codon-optimized four tandem repeats of FKBP (FKBP4) containing the F36V mutation, which allows it to bind to AP1903[40] was synthesized by Integrated DNA Technologies (IDT); this cDNA was used as a template for subcloning FKBP1-4 fusions for various constructs in Fig. 2 into pEGFP-N1. cDNA for Sp110 CARD (aa 6–110) were amplified from MegaMan Human Transcriptome Library (Stratagene). cDNA for the tandem SUMO interaction motifs of RNF4 (aa 38–129) was a gift from Xiaolu Yang (Addgene plasmid # 59743). Human SUMO1 and SUMO2 cDNA were generous gifts from Dr. A.D. Sharrocks (Manchester University; Manchester, United Kingdom). HA-SUMO constructs were subcloned into pCDNA3.1, hAire aa 1–106 and mAire aa 1–105 variants were subcloned into pET47 and pET50, respectively. Lentiviral vectors pMD2.G (encoding VSV-G), psPAX2 (encoding HIV Gag/Gag-Pol, Rev, and Tat), and pInducer20-GFP (encoding both tetracycline repressor and GFP downstream of a tetracycline response element) were generous gifts from Dr. Hidde Ploegh (Boston Children's Hospital; Boston, MA). DNA encoding GFP from pInducer20-GFP was replaced with hAire-FLAG variant DNA constructs. The vector pEQ276 (encoding CMV IE1 and 2 genes) was a gift from Adam Geballe (Addgene plasmid #83945). cDNA

of full-length IE1 was obtained from 293 T cells transiently transfected with pEQ276 (see below for transfection and cDNA generation methods) and subcloned with an N-terminal HA-tag into pCDNA 3.1. Please refer to Supplementary Table 1 for primers used for cloning.

**E. coli. expression and purification of Aire CARD**. His₆-3C-hAire CARD variants or His₆-NusA-His₆-3C-mAire CARD variants were co-expressed with pCDF-GroEL/ES + trigger factor (a generous gift from Timothy A. Springer lab, Boston Children's Hospital; Boston, MA) in BL21(DE3) cells at 37 °C. mAire CARD had to be expressed with a NusA-tag to increase solubility and prevent non-specific aggregation with *E. coli.* contaminants. Cells were grown to $OD_{600}$ ~0.25, then cooled down to 20 °C while still shaking for 45 min and induced with 0.4 mM IPTG at 20 °C for 16 h. Due to the toxicity of mAire CARD R70A/R71A, cells expressing this variant were instead cooled down to 25 °C and induced with 0.4 mM IPTG for 4 h. All cells were harvested, resuspended in CHAPS lysis buffer (20 mM HEPES pH 7.5, 250 mM NaCl, 10% glycerol, 20 mM imidazole, 0.05% CHAPS, 1 mM PMSF), and frozen at −20 °C. Thawed cell pellets were lysed with an Emulsiflex C3 (Avestin), and centrifuged at $32,000 \times g$ for 30 min at 4 °C. Cleared lysate was loaded onto a Ni²⁺-NTA agarose (Qiagen) gravity-flow column. The Ni²⁺-NTA column was washed with 100 column volumes of Wash Buffer (20 mM HEPES 7.5, 1 M KCl, 50 mM imidazole) and purified protein was eluted with 20 mM HEPES 7.5, 1 M KCl, 150–500 mM imidazole. In order for mAire CARD variants to form filaments, the His₆-NusA-3C-tag was cleaved off with the addition of His₆-MBP-tagged HRV 3 C protease [purified in-house by Ni²⁺-affinity, adding 1:10 ratio of protein mass] incubated at 4 °C for 16 h.

hAire CARD variants needed to be further purified by denaturation of the imidazole elutions in 6 M GdHCl at 37 °C for 30 min, followed by size exclusion chromatography with a Superdex 200 Increase 10/300 column (GE Healthcare) in 20 mM HEPES 7.5, 150 mM KCl, 6 M GdHCl, 5 mM βME. Purified fractions were pooled and snap-frozen. To refold the denatured CARD proteins, thawed protein was supplemented with 20 mM βME and dialyzed in a Slide-A-Lyzer MINI Dialysis Unit 7000 MWCO (Thermo Scientific) in 20 mM HEPES pH 7.5, 1 M KCl, 5 mM βME for 16 h at 4 °C. For hAire CARD variants, refolded proteins were recovered and incubated at 4 °C for another 24 h in order for filaments to form. For more in-depth biochemical analyses in Figs. 1b–d, 3C-protease-digested and untagged WT mAire CARD was denatured, further purified, and refolded in the same way.

**Negative-stain electron microscopy**. Carbon-coated hexagonal mesh copper grids (Electron Microscopy Sciences) were negatively glow-discharged for 30–45 s at 15 mA. Purified recombinant hAire and mAire CARD variants were diluted in TNE buffer (100 mM Tris pH 8, 50 mM NaCl, 10 mM EDTA) to a final concentration of ~50–200 μg/ml and immediately applied to the glow-discharged grid (Electron Microscopy Sciences). After 30 s, sample was blotted away with Whatman paper and washed with water twice. The grid was then pre-stained with 0.75% uranyl formate (UF) for 5 s, blotted and stained with 0.75% UF for 30 s. Stained grids were blotted and aspirated dry. Grids were imaged using a FEI Tecnai G² Spirit BioTWIN transmission electron microscope (TEM). In order to obtain higher resolution images of refolded mAire CARD wild-type filaments, grids were imaged with a FEI Tecnai F20 TEM. Images collected on the F20 TEM were analyzed with the image-processing suite EMAN2[67] and 2D-classification of the processed images was done with RELION 3.0[68].

**Congo Red binding and Thioflavin T fluorescence studies**. Congo Red (CR) binding and Thioflavin T (ThT) fluorescence studies were performed using a Spectra Max M5 plate reader (Molecular Devices). Briefly, CR and ThT dyes were dissolved in 20 mM Tris pH 7.5 and 0.22 μm filtered. For CR binding, proteins diluted into Amyloid Assay (AA) buffer (20 mM Tris pH 7.5, 110 mM NaCl, 250 mM KCl, 3 mM βME) were mixed with CR then incubated at 25 °C for 1 h. Samples were cleared by centrifugation at $18,000 \times g$ for 5 min and the absorbance from 430–600 nm for these samples were measured. For ThT fluorescence, proteins diluted in AA buffer were mixed with ThT. Fluorescence measurements were made at an excitation wavelength of 430 nm. The RIP1/3 RHIM complex, which is a bona fide amyloid and used as a positive control, was expressed (expression plasmids were a generous gift from Dr. Hao Wu, Boston Children's Hospital; Boston, MA) and purified by Ni²⁺-affinity and gel filtration chromatography.

**Cell culture and transfection**. 293 T cells (generous gift from Dr. Dan Stetson, University of Washington; Seattle, WA) were maintained in DMEM supplemented with 10% FBS, 1% L-glutamine. 293 T cells were transfected with either poly-ethyleneimine (PEI, 3.75 μg per well of 6-well plate with 1.5 μg DNA) or Lipofectamine 2000 (Invitrogen, 1–1.25 μg DNA per well of 12-well plate) according to manufacturer's protocol. The 4D6 cell line, originally derived from human thymic epithelium from children undergoing cardiac surgery, was maintained in RPMI supplemented with 10% FBS, 1% L-glutamine, and transfected with Lipofectamine 2000 (1–1.25 μg DNA per well of 12-well plate). The authenticity of 293 T cells was not verified. Both 293 T and 4D6 cell lines were verified to be mycoplasma free by using MycoAlert Mycoplasma Detection Kit (Lonza, Cat. No LT07–318).

For stable 4D6 cell line generation, lentivirus was first produced in 293 T cells. 293 T cells were seeded in a 6-well plate format and each well was transfected with

1.5 μg pInducer, 0.65 μg psPAX2, and 0.35 μg pMD2.G using Lipofectamine 2000. 16 h later, the medium was replaced. 48 h after transfection, the medium (inoculum) was harvested and passed through a 0.45 μm filter. For transduction, 4D6 cells were seeded in a 6-well plate. When cells were 70% confluent, the medium was replaced with a mixture of 500 μl filtered inoculum + 1.5 ml RPMI (supplemented with 10% FBS, 1% L-glutamine) + polybrene (1 mg/ml final concentration, Sigma-Aldrich). 24–30 h later, the transduced cells were trypsinized and transferred into 10 cm² dishes for selection in 1 mg/ml G418 sulfate (Corning). During G418 selection, medium was changed every 2–3 days. After the mock-transduced cells were ~95% dead, cells undergoing G418 selection were diluted into 96-well plates for individual clone selection.

**Antibodies**. Antibodies used for immunofluorescence (IF) microscopy were mouse anti-FLAG (M2, Sigma-Aldrich Cat. No. F3165), and rabbit anti-PML H238 (Santa Cruz Biotechnology Cat. No. sc5621); Cy5-, Alexa647-, or Alex488-conjugated anti-mouse IgG (Jackson ImmunoResearch Cat. No. 715–175–151, 715–605–151, 715–545–151); Cy3-, Cy5-, FITC-, or Alexa488-conjugated anti-rabbit IgG (Jackson ImmunoResearch Cat. No. 711–165–152, 711–175–152, 711–095–152, 711–545–152). All antibodies for IF microscopy were diluted 1:100. Antibodies used for immunoblotting were mouse anti-FLAG HRP (M2, Sigma-Aldrich Cat. No. A8592); mouse anti-histone H3 1B1B2 (Cell Signaling Technology Cat. No 2367S), rabbit anti-HA C29F4 (Cell Signaling Technology Cat. No. 3724S), anti-rabbit IgG-HRP (Cell Signaling Technology Cat. No. 7074S); anti-mouse IgG-HRP (GE Healthcare Cat. No NA931V). Primary antibodies for immunoblotting were diluted 1:1000–1:2500; HRP-conjugated antibodies were diluted 1:5000–1:10,000.

**Immunofluorescence microscopy**. 293 T or 4D6 cells were seeded onto poly-Lys-coated cover slips in 12-well plate format. Cells at ~70% confluence were transiently transfected with indicated plasmids. 4D6 stable cell lines expressing hAire under an inducible promoter were seeded in the presence of 1 μg/ml doxycycline (Fisher Scientific). For MG132 treatment, 4D6 cells that had been transfected ~20 h prior were incubated with 10 μm MG132 (Selleckchem) in RPMI supplemented with 1% FBS for 4 h. After 24 h post-transfection or doxycycline treatment, cells were washed with PBS, then fixed with 2% paraformaldehyde in PBS for 10 min. Cells were washed again with PBS and then permeabilized with 0.5% Triton X-100 in PBS for 10 min. Cells were blocked with 1% BSA in PBST (PBS + 0.2% Tween-20) for 15 min, and then probed with antibodies. Cells were then counterstained with DAPI (Life Technologies). Coverslips were mounted using Fluoromount-G (SouthernBiotech) and imaged with a wide-field Zeiss Axio Imager M1 fluorescence microscope using the software Slidebook 4.2.

An in-house MATLAB program (version R2018b) was used for automated image analyses. For each 2D image, masks were drawn around all nuclei based on DAPI fluorescence. Foci spot detection was carried out on a nucleus-by-nucleus basis. Within each nucleus, we first applied top-hat filtering to smooth and reduce the background noise. Next, we used a threshold (per nucleus) defined by the sum of mean and at least one standard deviation of the pixel intensities to outline the boundary of the foci[69]. Nuclei that contained fluorescent foci for both Aire and PML were examined further. The number of shared pixels for each Aire/PML foci pair was calculated, then divided by the number of pixels within the smallest focus of the pair, yielding a colocalized area percentage. Foci pairs with a colocalized area percentage greater than a threshold of 50% were defined as colocalizing. To test the robustness of our results, we also carried out this analysis with different thresholds ranging from 10 to 80% and we found that the qualitative results were the same. Statistical significance comparisons were calculated using a Student's *t*-test for two population proportions where each population consists of all individual Aire foci examined.

**Quantitative PCR**. 293 T, 4D6 cells or 4D6 stable cells + 0.2–1 μg/ml doxycycline were seeded in a 12-well plate format; cells were harvested 48 h post-transfection or post-induction, respectively. Total RNA was isolated using Direct-zol RNA mini prep kit (Zymo Research) and reverse-transcribed using SuperScript II (Life Technologies) with oligo(dT₁₈). qPCR was performed using Power SYBR Green PCR Master Mix (Invitrogen) on a CFX-Connect detection system (Bio-Rad, Hercules, CA) with Bio-Rad CFX Manager 3.1 software. The expression of Aire-induced genes was normalized against that of the Aire-independent gene *GAPDH* using the ΔΔCt method. *HPRT1* (Aire-independent gene control) was also normalized against *GAPDH*. The qPCR primer sequences are listed in Supplementary Table 2.

For MG132 and AP1903 treatments, after 16 h of transient transfection, the cell media was replaced with DMEM supplemented with 1% FBS, 1% L-glutamine with DMSO, 10 μM MG132 or 5 μM AP1903 (MedChemExpress). For AP1903-treated cells, 3–4 h later, 9% more FBS was supplemented into the media. Cells were harvested 16 h after MG132 or AP1903 treatments.

**Aire protein expression and chromatin fractionation assays**. 4D6 and 293 T cells were transfected with plasmids expressing indicated proteins for assaying expression levels and chromatin fractionation assays in 12-well and 6-well plate formats, respectively. Stable 4D6 cells induced with 0.2–1 μg/ml doxycycline were also seeded in 12-well format for determining expression levels. 48 h after

transfection or doxycycline induction, cells were harvested in PBS and washed one more time with PBS. Cells were incubated in hypotonic buffer [20 mM HEPES pH 7.5, 0.05% IGEPAL, 1.5 mM MgCl₂, 10 mM KCl, 5 mM EDTA, and 1X mammalian protease inhibitor cocktail (G-Biosciences); 50 μl and 100 μl/sample for 12-well and 6-well plate formats, respectively] for 15 min at 4 °C. The lysed cells were spun down at $500 \times g$ for 5 min at 4 °C and the supernatant (cytoplasmic fraction) was removed. The pellet (nuclear fraction) was washed two times with ice-cold PBS.

For the comparison of Aire variant expression levels, the PBS-washed nuclear fraction was lysed in 1% SDS Buffer (50 mM Tris pH 7.5, 150 mM NaCl, 1% SDS, 0.3 mM DTT; 75–100 μl/sample) and boiled for 15 min. BCA assay was used to determine the total protein concentration of lysates. Equal amounts of total protein from nuclear lysates were loaded on SDS-PAGE gel and subsequently analyzed by western blotting.

For the Aire chromatin fractionation assay, the PBS-washed nuclear fraction was resuspended in Nuclear Extraction buffer (50 mM Bis-Tris pH 7.5, 750 mM 6-aminocaproic acid, 3 mM CaCl₂, 10% glycerol, 1X mammalian protease inhibitor cocktail; 200 μl/sample) and split into fractions with or without MNase (Promega, 50 U/100 μl of nuclear lysate) and incubated for 1 h at 4 °C. MNase activity was quenched with 5 mM EDTA. Nuclear lysates were centrifuged at $18,000 \times g$ for 10 min at 4 °C. The resulting supernatant was the soluble nuclear fraction and saved for analysis. The insoluble nuclear pellet was washed one time with ice-cold PBS, then resuspended in Laemmli sample buffer and boiled for 5 min. The soluble and insoluble nuclear fractions were run on SDS-PAGE gel and subsequently analyzed by western blotting. Western blot quantification for chromatin fractionation assays was performed in ImageJ v.2.0.0

**Immunoprecipitation of Aire-FLAG**. 293 T cells were transiently transfected with Aire-FLAG variants and HA-SUMO expression vectors in a 12-well plate format. 24 h after transfection, media was replaced with DMEM supplemented with 1% FBS, 1% L-glutamine, and DMSO or 10 μM MG132. 48 h after transfection, cells were harvested in PBS, washed one more time in PBS, and lysed in 1% SDS buffer (100 μl/sample) and boiled for 15 min. Total cell lysates were cleared at $18,000 \times g$ for 10 min, then diluted 10-fold into Pull-down buffer (50 mM Tris pH 7.5, 150 mM NaCl, 1 mM EDTA, 1% TritonX-100). The diluted lysates were immuno-precipitated with anti-FLAG M2 affinity agarose beads (Sigma-Aldrich) at 4 °C for 4–16 h. Beads were washed two times with ice-cold PBS + 0.05% IGEPAL and 1 time with ice-cold PBS. To elute immunoprecipitated proteins, beads were boiled with Laemmli sample buffer for 5 min. Input samples and immunoprecipitated proteins were run on SDS-PAGE gel and analyzed by western blotting.

**Statistics and reproducibility**. All experiments were performed 2–4 independent times and were reproducible. Detailed methods of statistical analyses (error bar and p-value analysis) were indicated in figure legends and their values are provided in Source Data file.

**Reporting summary**. Further information on research design is available in the Nature Research Reporting Summary linked to this article.

## Data availability
The source data underlying Table 1, Fig. 1c, d, g, h; 2b; 3a-c, e; 4b-d, f; 5b, d; 6b-e; 7a, b, d and Supplementary Figures 1b, c, e, g; 2a, b; 3b, c; 4a, d-g; 5a, b; 6a, c-e and 7d, e are provided as a Source Data file. All other data are included in the Supplementary Information or available from the authors upon reasonable requests.

## Code availability
The in-house MATLAB program used for imaging analyses was specifically written for images acquired by Zeiss Axio Imager M1 fluorescence microscope. This code is available upon request.

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

## Acknowledgements

This work was supported by NIH T32 fellowship AI007512 (Y.H.), Burroughs Wellcome Fund (S.H.) and NIH grants (R35GM119537 to W.P.W., R01AI088204 to D.M., R01AI111784, and R21AI147099 to S.H.).

## Author contributions

Y.H., B.W., S.P., K.B., and E.G. performed the experiments. Y.H. and D.Y. performed the fluroscence microscopy image analyses. Y.H., W.P.W., D.M., and S.H. designed and coordinated the study. Y.H. and S.H. interpreted the results and wrote the manuscript. All authors discussed the results and commented on the manuscript.

## Competing interests
The authors declare no competing interests.
