## [Peer Review File · Nature Communications]

Editorial Note: This manuscript has been previously reviewed at another journal that is not operating a transparent peer review scheme. This document only contains reviewer comments and rebuttal letters for versions considered at Nature Communications .

Reviewers' comments:

Reviewer #1 (UPR-related pathology; Nature Ref#1)(Remarks to the Author):

The authors have not addressed my previous concern, which in my view is essential to place their findings in a disease-relevant context. Without these experiments, the disease relevance continues to be in doubt:

All the experiments are performed in rather standard cell lines that are unlikely to be relevant to APS-1. Although disease mechanism delineated here is plausible, there appears to be no validation of this mechanism in the disease itself. It would thus add enormously to this study if the authors could show that Aire is mislocalized to PML bodies in mTECs of patients as predicted by their model. Alternatively, mTECs could be generated from iPSCs from APS-1 patients and control subjects as described by Isobe and colleagues (e.g. DOI: 10.1038/icb.2010.96 and DOI: 10.1089/cell.2015.0006). In this way, it could also be tested whether APS-1-linked Aire variants colocalize with PML bodies in a disease-relevant setting.

Reviewer #2 (Thymocyte, histone remodeling; Nature Ref#2)(Remarks to the Author):

The authors have responded carefully to the reviewer's suggestions and the manuscript is much improved.

An important issue remains: The authors have not shown the in vivo relevance of their findings. Aire is not expressed in the cell lines they use to examine it and this is a serious limitation to their conclusions. While I feel the manuscript could be accepted the authors must express the limitations of their work. They have not show filament formation by Aire in mTEC cells and none of the studies were done in an appropriate cell type. Instead they used cells that vastly overexpress proteins and have been the source of many artifacts, like Hela cells.

The authors need to be very transparent about the limitations of their conclusions and their in vivo relevance to immune tolerance. Their studies could be relevant and interesting, but it is not clear.

Reviewer #3 (TCR signaling, Akt/mTOR; new Ref mediating for Nature Ref#3)(Remarks to the Author):

The authors have gone to great lengths here to respond to the previous critiques, performing additional quantitation and, when appropriate, softening conclusions in the text. The resulting manuscript appears to be much stronger at this point. Although questions remain about the relationship of CARD-mediated filaments to Aire function, this study should be of great interest to the field.

Reviewer #4 (Autoimmunity, Aire; Nature Ref#4)(Remarks to the Author):

Huoh et al report on AIRE multimerization and its role in transcriptional regulation. The authors' show

that AIRE CARD domain is needed to form filaments. They then study the location and transcriptional activation of AIRE protein and its several mutations. The authors also show that CARD can be replaced by FK506 binding protein domains that induced oligomerization and maintain some transcriptional activity. The study suggests that one main mechanism how AIRE mutants are inactive is that they are recruited to PML bodies. They show that mouse CARD mutant (like double mutant K53A/E54A) indeed are more SUMOylated and has co-localize with PML bodies, however, this is only partial colocalization. Also some disease-related mutations in PHD1 domain have partial overlap with PML bodies. The authors conclude that the mutations in AIRE lead to PML body colocalization and this is the reason for decreased transcriptional activity.

The work has some new findings such as the characterization of AIRE formed filaments and their structure. They also show that human AIRE CARD domain mutations L13P, T16M, L28P, A58V and K83E disrupt the filaments, which is a new finding. Some other aspects such as CARD-mediated multimerization and mutations-related decrease in transcriptional activity have been reported elsewhere by earlier studies. The main new claim the authors make is that the decrease of transcriptional activity is that AIRE mutants locate from AIRE bodies to PML bodies, which are known sites of transcriptional repression. However, the link between transcriptional downregulation and PML body location is not solid and neither is it strong enough to generalize it as key outcome of AIRE mutations.

The work mostly focus on induced double mutation in CARD domain of mouse AIRE gene - K53A/E54A (and PML targeting SUMO interaction motif (SIM) construct). However, they provide little evidence for human AIRE CARD mutations. At least 20 different missense mutations in AIRE CARD domain have been identified, and many of them have been studied earlier for transcriptional activity and subcellular localizations. These reports have showed a variable degree of intracellular locations and varying effects on transcriptional activities. The colocalization of PHD domain mutations in Figure 6c seems to be partial C302Y (25%) and C311Y (10%), and it remains unclear whether the complete downregulation of AIRE transcriptional activity can be explained by this limited colocalization. Also in Figure 7b, it seems that when the dominant PHD1 mutations are coexpressed with WT AIRE coexpressed, most of the bodies are still outside of PML bodies and not overlapping. Finally, the authors should consider that the filaments they describe and which are disrupted by CARD mutations are not related to nuclear bodies as AIRE is well known to also form cytoplasmic filaments when transfected into the cells. These filaments in cytoplasm are not necessarily related to the transcriptional activity.

To make a convincing case, several single human AIRE CARD and PHD1/PHD2 mutations need to be studied to prove this as a general mechanism and not just related to few mutations with relatively modest overlap. Furthermore the authors should provide a more mechanistic protein link between the decreased AIRE transcriptional activity and recruitment to PML bodies. Equally important would be to show the colocalization with PML bodies in vivo situations.

Point-by-Point rebuttal

Reviewer #1 (UPR-related pathology; Nature Ref#1)(Remarks to the Author):

The authors have not addressed my previous concern, which in my view is essential to place their findings in a disease-relevant context. Without these experiments, the disease relevance continues to be in doubt:

All the experiments are performed in rather standard cell lines that are unlikely to be relevant to APS-1. Although disease mechanism delineated here is plausible, there appears to be no validation of this mechanism in the disease itself. It would thus add enormously to this study if the authors could show that Aire is mislocalized to PML bodies in mTECs of patients as predicted by their model. Alternatively, mTECs could be generated from iPSCs from APS-1 patients and control subjects as described by Isobe and colleagues (e.g. DOI: 10.1038/icb.2010.96 and DOI: 10.1089/cell.2015.0006). In this way, it could also be tested whether APS-1-linked Aire variants colocalize with PML bodies in a disease-relevant setting.

> We acknowledge the limitation of our study, which is now explicitly stated in Discussion.

Reviewer #2 (Thymocyte, histone remodeling; Nature Ref#2)(Remarks to the Author):

The authors have responded carefully to the reviewer's suggestions and the manuscript is much improved.

An important issue remains: The authors have not shown the in vivo relevance of their findings. Aire is not expressed in the cell lines they use to examine it and this is a serious limitation to their conclusions. While I feel the manuscript could be accepted the authors must express the limitations of their work. They have not show filament formation by Aire in mTEC cells and none of the studies were done in an appropriate cell type. Instead they used cells that vastly overexpress proteins and have been the source of many artifacts, like Hela cells.

The authors need to be very transparent about the limitations of their conclusions and their in vivo relevance to immune tolerance. Their studies could be relevant and interesting, but it is not clear.

> We acknowledge the limitation of our study. The main text was edited throughout the manuscript to clearly lay out the limitation of our study (colored red). We also added explicit statements in Discussion to address the reviewer's concern.

Reviewer #3 (TCR signaling, Akt/mTOR; new Ref mediating for Nature Ref#3)(Remarks to the Author):

The authors have gone to great lengths here to respond to the previous critiques, performing additional quantitation and, when appropriate, softening conclusions in the text. The resulting manuscript appears to be much stronger at this point. Although questions remain about the relationship of CARD-mediated filaments to Aire function, this study should be of great interest to the field.

> We thank the reviewer for the positive comment.

Reviewer #4 (Autoimmunity, Aire; Nature Ref#4)(Remarks to the Author):

Huoh et al report on AIRE multimerization and its role in transcriptional regulation. The authors' show that AIRE CARD domain is needed to form filaments. They then study the location and transcriptional activation of AIRE protein and its several mutations. The authors also show that CARD can be replaced by FK506 binding protein domains that induced oligomerization and maintain some transcriptional activity. The study suggests that one main mechanism how AIRE mutants are inactive is that they are recruited to PML bodies. They show that mouse CARD mutant (like double mutant K53A/E54A) indeed are more SUMOylated and has co-localize with PML bodies, however, this is only partial colocalization. Also some disease-related mutations in PHD1 domain have partial overlap with PML bodies. The authors conclude that the mutations in AIRE lead to PML body colocalization and this is the reason for decreased transcriptional activity.

The work has some new findings such as the characterization of AIRE formed filaments and their structure. They also show that human AIRE CARD domain mutations L13P, T16M, L28P, A58V and K83E disrupt the filaments, which is a new finding. Some other aspects such as CARD-mediated multimerization and mutations-related decrease in transcriptional activity have been reported elsewhere by earlier studies. The main new claim the authors make is that the decrease of transcriptional activity is that AIRE mutants locate from AIRE bodies to PML bodies, which are known sites of transcriptional repression. However, the link between transcriptional downregulation and PML body location is not solid and neither is it strong enough to generalize it as key outcome of AIRE mutations.

The work mostly focus on induced double mutation in CARD domain of mouse AIRE gene - K53A/E54A (and PML targeting SUMO interaction motif (SIM) construct). However, they provide little evidence for human AIRE CARD mutations. At least 20 different missense mutations in AIRE CARD domain have been identified, and many

of them have been studied earlier for transcriptional activity and subcellular localizations. These reports have showed a variable degree of intracellular locations and varying effects on transcriptional activities. The colocalization of PHD domain mutations in Figure 6c seems to be partial C302Y (25%) and C311Y (10%), and it remains unclear whether the complete downregulation of AIRE transcriptional activity can be explained by this limited colocalization. Also in Figure 7b, it seems that when the dominant PHD1 mutations are coexpressed with WT AIRE coexpressed, most of the bodies are still outside of PML bodies and not overlapping. Finally, the authors should consider that the filaments they describe and which are disrupted by CARD mutations are not related to nuclear bodies as AIRE is well known to also form cytoplasmic filaments when transfected into the cells. These filaments in cytoplasm are not necessarily related to the transcriptional activity.

To make a convincing case, several single human AIRE CARD and PHD1/PHD2 mutations need to be studied to prove this as a general mechanism and not just related to few mutations with relatively modest overlap. Furthermore the authors should provide a more mechanistic protein link between the decreased AIRE transcriptional activity and recruitment to PML bodies. Equally important would be to show the colocalization with PML bodies in vivo situations.

>In the revised manuscript, we now include our analysis of additional APS-1 mutations. Together with the mutations presented in the previous version of the manuscript, this amounts to 14 APS-1 mutations: 6 mutations in CARD, 3 in SAND (including 1 premature stop mutation) and 5 in PHD1 (Table 1). Based on this expanded analysis, we agree with the reviewer in that most CARD mutations impair the foci forming activity, and thus do not co-localize with PML bodies. However, we note that this result does not argue against our conclusion that CARD can affect PML localization of Aire foci, as we found several examples from our protein engineering approach that support this model. Unlike the CARD mutations, we found that all SAND and PHD1 mutations we tested do maintain foci formation. Importantly, more than half of them (5 out of 8) in fact show increased PML localization and a dominant negative effect in the transcriptional activity assay. While we also found that not all dominant negative mutants display increased PML association, together with our mechanistic analyses in Figs 4-7, our observations suggest that PML co-localization is a common feature frequently associated with dominant negative activity. This new piece of data is now presented in Supplementary Fig. 6b-c and summarized in Table 1.

We also acknowledge the limitation of our study as the reviewer pointed out. As such, the main text was edited throughout the manuscript to clearly lay out the lack of *in vivo* and mTEC analysis (colored red). We also added explicit statements in Discussion to discuss this issue, and future direction.

REVIEWERS' COMMENTS:

Reviewer #2 (Remarks to the Author):

The authors have responded effectively by selective rewriting to demonstrate the limitations of their conclusions.

Reviewer #4 (Remarks to the Author):

Overall, the revised version has improved. However, the authors seem to ignore, but should critically note in their paper, that the AIRE protein is well-known in the field to form cytoplasmic filamentous structures, when transfected into tissue culture cells, which has been reported by many papers on AIRE protein. From current results reported by Huoh et al, it is difficult to distinguish whether the filaments seen in the transfected cell lines represent nuclear bodies or are cytoplasmic filaments located in cytoplasm. Papers that have showed AIRE subcellular localization in cytoplasmic filaments (in addition to its nuclear location) are:

Rinderle C, Christensen HM, Schweiger S, Lehrach H, Yaspo ML. (1999) AIRE encodes a nuclear protein co-localizing with cytoskeletal filaments: altered sub-cellular distribution of mutants lacking the PHD zinc fingers. *Hum Mol Genet.* 8: 277-90

Heino M, Peterson P, Kudoh J, Nagamine K, Lagerstedt A, Ovod V, Ranki A, Rantala I, Nieminen M, Tuukkanen J, Scott HS, Antonarakis SE, Shimizu N, Krohn K. (1999) Autoimmune regulator is expressed in the cells regulating immune tolerance in thymus medulla. *Biochem Biophys Res Commun.* 257: 821-5.

Björnses P, Pelto-Huikko M, Kaukonen J, Aaltonen J, Peltonen L, Ulmanen I. (1999) Localization of the APECED protein in distinct nuclear structures. *Hum Mol Genet.* 1999 8: 259-66.

Björnses P, Halonen M, Palvimo JJ, Kolmer M, Aaltonen J, Ellonen P, Perheentupa J, Ulmanen I, Peltonen L. (2000) Mutations in the AIRE gene: effects on subcellular location and transactivation function of the autoimmune polyendocrinopathy-candidiasis-ectodermal dystrophy protein. *Am J Hum Genet.* 66: 378-92.

Pitkänen J, Vähämurto P, Krohn K, Peterson P. (2001) Subcellular localization of the autoimmune regulator protein. characterization of nuclear targeting and transcriptional activation domain. *J Biol Chem.* 276: 19597-602.

Ramsey C, Bukrinsky A, Peltonen L. (2002) Systematic mutagenesis of the functional domains of AIRE reveals their role in intracellular targeting. *Hum Mol Genet.* 11: 3299-308.

Halonen M, Kangas H, Ruppell T, Ilmarinen T, Ollila J, Kolmer M, Vihinen M, Palvimo J, Saarela J, Ulmanen I, Eskelin P. (2004) APECED-causing mutations in AIRE reveal the functional domains of the protein. *Hum Mutat.* 23: 245-57.

Ilmarinen T, Melén K, Kangas H, Julkunen I, Ulmanen I, Eskelin P. (2006) The monopartite nuclear localization signal of autoimmune regulator mediates its nuclear import and interaction with multiple importin alpha molecules. *FEBS J.* 273: 315-24.

REVIEWERS' COMMENTS:

Reviewer #4 (Remarks to the Author):

Overall, the revised version has improved. However, the authors seem to ignore, but should critically note in their paper, that the AIRE protein is well-known in the field to form cytoplasmic filamentous structures, when transfected into tissue culture cells, which has been reported by many papers on AIRE protein. From current results reported by Huoh et al, it is difficult to distinguish whether the filaments seen in the transfected cell lines represent nuclear bodies or are cytoplasmic filaments located in cytoplasm. Papers that have showed AIRE subcellular localization in cytoplasmic filaments (in addition to its nuclear location) are:

Rinderle C, Christensen HM, Schweiger S, Lehrach H, Yaspo ML. (1999) AIRE encodes a nuclear protein co-localizing with cytoskeletal filaments: altered sub-cellular distribution of mutants lacking the PHD zinc fingers. *Hum Mol Genet.* 8: 277-90

Heino M, Peterson P, Kudoh J, Nagamine K, Lagerstedt A, Ovod V, Ranki A, Rantala I, Nieminen M, Tuukkanen J, Scott HS, Antonarakis SE, Shimizu N, Krohn K. (1999) Autoimmune regulator is expressed in the cells regulating immune tolerance in thymus medulla. *Biochem Biophys Res Commun.* 257: 821-5.

Björnses P, Peltö-Huikko M, Kaukonen J, Aaltonen J, Peltonen L, Ulmanen I. (1999) Localization of the APECED protein in distinct nuclear structures. *Hum Mol Genet.* 1999 8: 259-66.

Björnses P, Halonen M, Palvimo JJ, Kolmer M, Aaltonen J, Ellonen P, Perheentupa J, Ulmanen I, Peltonen L. (2000) Mutations in the AIRE gene: effects on subcellular location and transactivation function of the autoimmune polyendocrinopathy-candidiasis-ectodermal dystrophy protein. *Am J Hum Genet.* 66: 378-92.

Pitkänen J, Vähämurto P, Krohn K, Peterson P. (2001) Subcellular localization of the autoimmune regulator protein. characterization of nuclear targeting and transcriptional activation domain. *J Biol Chem.* 276: 19597-602.

Ramsey C, Bukrinsky A, Peltonen L. (2002) Systematic mutagenesis of the functional domains of AIRE reveals their role in intracellular targeting. *Hum Mol Genet.* 11: 3299-308.

Halonen M, Kangas H, Rüppell T, Ilmarinen T, Ollila J, Kolmer M, Vihinen M, Palvimo J, Saarela J, Ulmanen I, Eskelin P. (2004) APECED-causing mutations in AIRE reveal the functional domains of the protein. *Hum Mutat.* 23: 245-57.

Ilmarinen T, Melén K, Kangas H, Julkunen I, Ulmanen I, Eskelin P. (2006) The monopartite nuclear localization signal of autoimmune regulator mediates its nuclear import and interaction with multiple importin alpha molecules. FEBS J. 273: 315-24.

We thank Reviewer #4 for the comment. We have included mention of this in the Discussion.